# Nuclear Delivery of Nanoparticle-Based Drug Delivery Systems by Nuclear Localization Signals

**DOI:** 10.3390/cells12121637

**Published:** 2023-06-15

**Authors:** Yuhan Nie, Guo Fu, Yuxin Leng

**Affiliations:** 1Innovation and Integration Center of New Laser Technology, Shanghai Institute of Optics and Fine Mechanics, Chinese Academy of Sciences, Shanghai 201800, China; yuhannie@mail.ustc.edu.cn; 2State Key Laboratory of High Field Laser Physics and CAS Center for Excellence in Ultra-Intense Laser Science, Shanghai Institute of Optics and Fine Mechanics, Chinese Academy of Sciences, Shanghai 201800, China

**Keywords:** nanomedicine 2.0, NPCs, NDDSs, nuclear delivery, NPs, NLS

## Abstract

Nanomedicine 2.0 refers to the next generation of nanotechnology-based medical therapies and diagnostic tools. This field focuses on the development of more sophisticated and precise nanoparticles (NPs) for targeted drug delivery, imaging, and sensing. It has been established that the nuclear delivery of NP-loaded drugs can increase their therapeutic efficacy. To effectively direct the NPs to the nucleus, the attachment of nuclear localization signals (NLSs) to NPs has been employed in many applications. In this review, we will provide an overview of the structure of nuclear pore complexes (NPCs) and the classic nuclear import mechanism. Additionally, we will explore various nanoparticles, including their synthesis, functionalization, drug loading and release mechanisms, nuclear targeting strategies, and potential applications. Finally, we will highlight the challenges associated with developing nucleus-targeted nanoparticle-based drug delivery systems (NDDSs) and provide insights into the future of NDDSs.

## 1. Introduction

Nanoparticle-based drug delivery systems (NDDSs) have revolutionized the medical field by providing targeted and efficient drug delivery while minimizing side effects [1]. Nuclear-targeted nanoparticles, in particular, have gained significant attention for their ability to deliver therapeutic drugs specifically to the nucleus of cancer cells. This is important because drugs delivered to the cytoplasm face challenges such as enzymatic degradation, acidic conditions, and being pumped out of cells by efflux transporters such as P-glycoprotein (P-gp) [2]. The cytosolic environment is crowded, comprising various macromolecules that can significantly impede the trafficking of macromolecules and nanoparticles. Hence, the mere internalization of a drug molecule into the cytosol does not guarantee its interaction with the desired subcellular target. Therefore, without specific design considerations, the ability of drugs to reach their targets can be greatly compromised. The nucleus, as the cell’s control center, plays a pivotal role in various disorders such as cancer, heart dysfunction, and brain disorders [3,4]. So, many drugs exert their primary effects within the nucleus [5,6]. Thus, the nuclear delivery of these drugs can improve their efficacy by increasing the effective concentration of drugs in the nucleus and avoiding multidrug resistance (MDR) caused by the overexpression of efflux transporters in the cytosol [7].

Nucleus-targeted nanoparticles (NPs) are designed and engineered to selectively accumulate in the nuclei of cancer cells, where they deliver drugs, genes, or other therapeutic drugs directly [8,9]. In order to achieve the targeted delivery of nanoparticles (NPs) to the nucleus, several requirements must be met. Firstly, NPs need to be labeled with nuclear localization signals (NLSs) to facilitate their nuclear targeting. Additionally, these NPs must overcome four distinct barriers during their journey: (1) traversing the cell membrane, (2) avoiding entrapment and degradation within endo/lysosomes, (3) achieving successful cytoplasmic trafficking, and, most importantly, (4) efficiently entering the nucleus. Overcoming these barriers is essential for ensuring the effective and specific delivery of NPs to the nucleus [6]. Hence, to improve the efficiency of nuclear entry, a good understanding of the NPC structure and mechanism of nuclear import is extremely important.

This review will first provide a brief introduction to the structure of NPCs and the mechanisms of classic and transactivator of transcription (TAT) nuclear import. We will then focus on various nanoparticles, covering topics such as their synthesis, functionalization, drug loading and release mechanisms, nuclear targeting strategies, and potential applications. Finally, we will discuss the challenges associated with nucleus-targeted NDDSs and provide perspectives on the future of NDDSs.

## 2. Structure of NPCs and Nuclear Import

### 2.1. The Structure of NPCs

The nuclear pore complex (NPC) is a large structure that exhibits octagonal rotational symmetry. In vertebrates, the NPC has a diameter of approximately 120 nm and a length of about 200 nm along the transport axis [10,11,12]. The transport axis is flanked by eight cytoplasmic filaments that are around 50 nm long. The central tube is hourglass-shaped, with a minimum diameter of approximately 50 nm and a length of around 85 nm [13]. The nuclear basket is 75 nm long and formed by eight filaments [12,14] (Figure 1). Perpendicular to the transport axis, the NPC can be divided into three layers: the lumenal ring, the core scaffold, and the phenylalanine–glycine (FG) nucleoporins. The lumenal ring is formed by the lumen domains of transmembrane proteins such as Pom121. The core scaffold consists of folded protein complexes and act as a scaffold to link the FG Nups to the nuclear envelope (Figure 1). The FG Nups are composed of approximately 200–250 intrinsically disordered polypeptides that form a permeability barrier, allowing ions and small molecules to pass but blocking large molecules. In the FG Nups layer, there are around 3000–4000 phenylalanine–glycine (FG) repeats. Nucleocytoplasmic transport receptors (NTRs) transiently bind to the FG repeats and facilitate the passage of carried cargo through the NPC [15,16,17,18,19].

### 2.2. Nuclear Import Pathways: Free Diffusion and Classical Nuclear Import Pathway

As is shown in Figure 2, to pass the NPCs, there are basic two pathways: one is free diffusion and the other is facilitated diffusion. For ions and small proteins (<40 kDa) [20], they can pass through the NPCs by free diffusion, exhibiting Brownian motion, and such molecules diffuse without a specific direction, whereas larger molecules travel from a high concentration to a low concentration. Free diffusion is a form of bidirectional diffusion, meaning that some molecules diffuse into the nucleus while some diffuse out of the nucleus. The upper size limit of molecules that can pass the NPCs by free diffusion is ~10 nm in diameter (~40 KDa).

The best-understood type of facilitated diffusion is the classical nuclear import pathway. Proteins destined for transport into the nucleus contain amino-acid-targeting sequences called nuclear localization signals (NLSs), which are recognized by importin α in the cytoplasm, linking it to importin β (importins are nuclear transport receptors responsible for importing cargo into nucleus). Importin β then mediates the interaction of the trimeric complex with the nuclear pore as it translocates into the nucleus. Once the import complex reaches the nucleus, it is dissociated by Ran-GTP. The binding of Ran-GTP to importin β causes a conformational change that results in the release of the importin α-cargo complex. The importin β will return the cytoplasma by itself and importin α will be exported by exportin CAS (exportins are nuclear transport receptors responsible for exporting cargo out of nucleus) [21,22]. The nuclear import of NLSs cargo occurs in a unidirectional manner due to the higher concentration of Ran-GTP in the nucleus compared to the cytoplasm. This asymmetry in Ran-GTP concentration ensures that the cargo is released exclusively on the nuclear side. In addition, cargo cannot return to the cytoplasm either by itself or by binding to importin α again in the nucleus. The maximum size of cargo that can be imported through facilitated diffusion is ~40 nm in diameter [23,24,25,26,27].

The classical nuclear import pathway offers distinct advantages compared to free diffusion. Firstly, it enables efficient and unidirectional transport, preventing the escape of cargo containing nuclear localization signals (NLSs) from the nucleus and promoting their accumulation within the nucleus. This unidirectional transport ensures that NLSs cargo remains confined to the nucleus, enhancing its functional effects. Secondly, the classical nuclear import pathway permits the entry of larger cargo molecules, as it can accommodate cargo with a diameter of up to 40 nm, whereas free diffusion is limited to smaller molecules with a diameter of around 10 nm. This ability allows for the nuclear import of larger macromolecules and facilitates their involvement in crucial nuclear processes.

### 2.3. Classical Nuclear Localization Signals (cNLSs) and Importin α

The cNLSs are widely considered as the prototype and most extensively studied form of NLSs. They were the initial NLSs to be discovered and characterized, leading to the identification of numerous proteins that utilize the classical import pathway. As a result, a wealth of examples have demonstrated the utilization of the cNLSs for nuclear import. The cNLSs typically consist of either one cluster (monopartite) or two clusters (bipartite) of basic residues. Monopartite NLSs, such as SV40 large-T antigen (PKKKRRV^132^), polyoma large T antigen (VSRKRPRP^196^), hepatitis D virus antigen (EGAPPAKRAR^75^), murine p53 (PPQPKKKPLDGE^322^), NF-κB p50 (QRKRQK^372^), NF-κB p65 (EEKRKR^286^), and human c-myc (PAAKRVKLD^328^/RQRRNELKRSF^374^) usually contain a single cluster of four or five basic residues. In contrast, bipartite NLSs, such as those found in Xenopus nucleoplasmin (KRPAATKKAGQAKKKKLD^171^), rat glucocorticoid receptor (YRKCLQAGMNLEARKTKKKIKGIQQATA^524^), and RCC1 (MSPKRIAKRRSPPADAIPKSKKVKVSHR^28^), have two clusters of basic residues. The first cluster is located in the usual position, but a second basic cluster can be found approximately 9–12 residues downstream of the first one [20,28].

Importin α is responsible for the nuclear import of proteins containing a classical nuclear localization signal. The central region of importin α contains a series of ten consecutive arginine-rich motif (ARM) repetitions, which collectively form a domain responsible for binding to the NLS. Structural studies have revealed the presence of two distinct NLS binding sites within the central domain. The first binding site, situated between the ARM motifs 1–4, directly interacts with the amino acids of monopartite NLSs as well as the longer amino acid sequence of bipartite NLSs. On the other hand, the second binding site, located in the ARM motifs 7–8, specifically binds to the shorter amino acid sequence found in bipartite NLSs. Structural analysis has demonstrated that mutations in the amino acids of the first NLS binding site severely impede the interaction between importin α and both types of cNLSs, and amino acid substitutions in the second NLS binding site decrease the affinity between importin α and bipartite NLSs, but do not affect the interaction with monopartite NLSs [29].

Experimental findings have revealed a direct correlation between the binding affinity of a classical nuclear localization signal (cNLS) for importin α and the nuclear accumulation and import rate of the corresponding cNLS cargo. A successful cNLS exhibits a binding constant of approximately 10 nM for importin α. Alternatively, a weak binding between importin α and the cNLS results in inefficient import of the cargo, while a tight binding hinders the effective release of the cNLS cargo from importin α within the nucleus. Computer simulations have further demonstrated that the import and accumulation of cNLS cargo in the nucleus are influenced by both the affinity of the cNLS cargo for importin α and the concentration of the importin α receptor [20].

The recognition of the cNLS by importin α is only the initial step in the formation of functional import complexes. The interaction between importin α and importin β is essential for an efficient transport process. This interaction is facilitated by the importin beta binding (IBB) domain located in the N-terminal region of importin α. Experimental evidence strongly supports a model in which proteins containing a cNLS have a higher affinity for importin α when it is associated with importin β, compared with binding to free importin β alone. Import complexes traverse the nuclear pore through a mechanism that involves the interaction between importin β and the FG domains of nucleoporins [29]. Further information regarding the NLS sequence and unconventional mechanisms of nuclear transport have been previously reviewed elsewhere [6,30,31,32].

### 2.4. Nuclear Import Pathways: Transactivator of Transcription (TAT) Pathway

The transactivator of transcription (TAT) peptide (^48^ GRKKRRQRRRAPQN ^61^), derived from human immunodeficiency virus-1 (HIV-1), has demonstrated remarkable capabilities in facilitating the delivery of various cargoes into cells and the nucleus. These cargoes range from nanoparticles, peptides, and nucleic acids to proteins. The TAT peptide contains overlapping regions responsible for cellular localization, including a classical nuclear localization signals (cNLSs) (^48^GRKKRR), and a cell-penetrating peptide (CPP) signal (^48^GRKKRRQRRRAPQN). These distinct regions within TAT play crucial roles in efficiently transporting cargoes across cellular barriers and facilitating their entry into the nucleus [33].

Despite the presence of a classical nuclear localization signals (cNLSs) within its sequence, the mechanism by which the TAT peptide is imported into the nucleus is still a topic of debate. Some evidence suggests that TAT does not interact with importin α but instead directly binds to importin β. In vitro nuclear import assays indicate that TAT’s nuclear entry is facilitated by its direct binding to importin β [34]. In a study by Efthymiadis et al., it was observed that TAT does not interact with importin α or importin β [35]. Surprisingly, despite the lack of interaction with these classical nuclear import receptors, TAT was found to effectively import a 476 kDa beta-galactosidase protein to the nucleus in an ATP-dependent manner, independent of cytosolic factors. These findings suggest that TAT utilizes ATP-dependent mechanisms to overcome cytoplasmic retention and facilitate nuclear entry. Furthermore, TAT was found to bind to nuclear components, potentially including RNAs. These results highlight the unique nuclear targeting abilities of TAT and suggest its involvement in interactions with nuclear molecules beyond the classical import pathways mediated by importin α and importin β [34,36]. A study conducted by Cardarelli et al. provided evidence supporting passive diffusion as the primary mechanism for TAT peptide-mediated nuclear transport in live cells [37]. Subsequent investigations have also supported these findings. In their study, Cardarelli et al. observed that TAT peptide, in the intact cellular environment, does not establish interactions with importin α or importin β. However, TAT does exhibit binding to importin α and importin β in the absence of intracellular cytosolic and nuclear factors. These results suggest that TAT remains inactive in the cellular context [38]. In one study, Nitin et al. made an interesting discovery regarding TAT peptide-based delivery [39]. They found that the transport mediated by TAT peptide is not affected by WGA blockage and does not rely on ATP. Surprisingly, TAT peptide demonstrated the ability to import 90 nm beads into the nuclei of digitonin-permeabilized cells. This suggests that the interaction of TAT peptide with the nuclear envelope follows a mechanism distinct from that of classical nuclear localization signals (cNLSs) [39]. Furthermore, it has been proposed that TAT peptide might enter the nucleus by directly crossing the nuclear envelope instead of utilizing the nuclear pore complex (NPC) [6]. According to a study conducted by Quan et al., it has been observed that TAT peptide exhibits limited membrane penetration at low concentrations. However, once the peptide concentration surpasses a threshold level, it can traverse the membrane by inducing nanopore formation, facilitated by the transmembrane electrostatic potential difference [40]. Nevertheless, it is important to note that there is currently no direct evidence demonstrating the formation of nanopores on the nuclear envelope by TAT peptide, enabling its entry into the nucleus. In contrast to earlier results, a recent report proposed that TAT peptide can enter the nucleus by interacting with importin α, without direct interaction with importin β. The study challenges previous findings and suggests an alternative mechanism for TAT peptide’s nuclear entry, highlighting the potential role of importin α in facilitating its translocation into the nucleus [33].

The nuclear import pathway of TAT peptide is illustrated in Figure 3. (1) Simple diffusion, no direct interaction with importin α and importin β. (2) Facilitated diffusion, direct interaction with importin α, but no direct interaction with importin β. (3) Facilitated diffusion, direct interaction with importin β, but no direct interaction with importin α. (4) Crossing the nuclear envelope directly. (5) Binding of nuclear components, such as RNA.

## 3. Nanoparticles for Nucleus Targeting

There are various types of nanoparticles that have been utilized as drug delivery systems (Figure 4), each with its own unique characteristics. Nanoparticles offer many advantages as drug delivery systems, including improved drug stability, increased efficacy, reduced cytotoxicity and side effects. Figure 4 shows the nanoparticle-based drug delivery systems (NDDSs) mainly used for nuclear targeted delivery of drugs. In the following sections, we will discuss nanoparticles that have been used to target the nucleus through nuclear localization signals (NLSs). We will provide an overview of their preparation and features, drug loading and release methods, nucleus targeting strategies, and applications.

### 3.1. Metal Nanoparticles

Metal nanoparticles are small particles made of various metals, such as gold, silver, platinum, iron, and copper, at the nanoscale level [41,42,43]. These nanoparticles exhibit unique physicochemical properties that differ from their bulk counterparts, making them attractive for a wide range of applications, including medicine, electronics, catalysis, and environmental science. Metal nanoparticles can serve as carriers for delivering therapeutic agents, such as drugs or genetic materials, to specific target sites in the body. Their small size, large surface area, and surface functionalization capabilities allow for the efficient encapsulation and controlled release of drugs, enhancing their efficacy and reducing side effects. Among them, gold nanoparticles (GNPs) have received significant attention and have been extensively studied as drug delivery vehicles.

Gold nanoparticles (GNPs) have shown great promise as effective carriers for drug delivery. Their negatively charged surfaces make them easily functionalize with various biomolecules such as drugs, genes, and targeting ligands, allowing them to transport multiple therapeutic agents to targeted sites. Additionally, GNPs offer unique physical and chemical properties that allow for controlled drug release through external stimuli such as light or internal stimuli such as pH or glutathione. GNPs are biocompatible, non-toxic, and safe for use in biological systems. They can be synthesized easily and produced in a variety of shapes and sizes, ranging from 1 nm to over 100 nm, including spherical, rod-like, cage-like, and other forms. The distinctive surface effects, ultra-small size, macroscopic quantum tunneling effect, and presence of surface plasmon resonance (SPR) bands make GNPs ideal for various biomedical applications, such as biosensing, molecular imaging, and drug delivery systems [44].

Spherical gold nanoparticles (GNPs) ranging from 1 to 100 nm are a versatile choice for drug delivery applications. They can be synthesized using various methods, including the conventional synthesis method, which involves reducing aqueous chloroauric acid with sodium citrate. In this process, citrate acts as a stabilizer and reducing agent, and the size of the resulting GNPs can be adjusted by altering the stoichiometric ratio between chloroauric acid and sodium citrate. Precise control of the synthesis process is crucial, since the size and shape of GNPs significantly impact their drug delivery efficiency. Alternative synthesis methods also exist for producing spherical GNPs, which can be selected based on specific application requirements.

The functionalization of gold nanoparticles (GNPs) with various biomolecules, such as DNA, peptides, and antibodies, is essential for their biomedical applications. These biomolecules can be attached to the surface of GNPs through either noncovalent interactions, such as electrostatic interactions, hydrophobic entrapment, and van der Waals forces, or covalent modifications, which involve immediate chemical attachment, linker molecules, or click chemistry [45]. Noncovalent interactions can be used to load drugs onto functionalized GNPs, which can be released by diffusion, low pH, chemical reactions, or physical stimuli. Alternatively, drugs can be covalently conjugated to GNPs through a cleavable linker and then released by light or glutathione [46].

Kang et al. conducted a study to investigate the interactions and effects of gold nanoparticles (GNPs) in living cells [39]. To accomplish this, the researchers synthesized 30 nm gold nanoparticles (GNPs) and coated them with polyethylene glycol (PEG). Then, these GNPs were further bioconjugated with RGD peptide and NLS peptide (KKKRK). RGD targets αvβ6 integrins on the cell surface, allowing it to enter the cytoplasm via receptor-mediated endocytosis. The conjugation of GNPs with RGD only (RGD-GNPs) resulted in the specific targeting of cancer cells, while conjugation with both RGD and NLS peptide (RGD/NLS-GNPs) resulted in the specific targeting of nuclei of cancer cells. The study used human oral squamous carcinoma cell (HSC) as a cancer cell model and human keratinocytes (HaCat) as a normal cell model. The RGD/NLS-GNPs specifically targeted the nuclei of cancer cells over those of normal cells. The study found that cytokinesis arrest occurred only in cancer cells treated with 0.4 nM RGD/NLS-GNPs, indicating that disruption within the nucleus was the cause. The study concludes that the nuclear targeting of gold nanoparticles in cancer cells can lead to cytokinesis arrest, resulting in apoptosis. These findings suggest that GNPs can be used as an anticancer therapeutic material if conjugated to the proper nuclear-targeting ligands.

In a study conducted by Peng et al., a simple method was employed to prepare cationic gold and silver nanoparticles functionalized with TAT peptides for gene delivery to epidermal stem cells [47] (Figure 5). Positively charged PEI molecules were utilized as capping agents to fabricate AuNPs and AgNPs (referred to as Au@PEI and Ag@PEI NPs, respectively), with diameters of 20 nm and 35 nm, respectively. Subsequently, the nanoparticles underwent partial ligand exchange with thiol-TAT peptides, resulting in the formation of NPs@PEI-TAT (TAT: RKKRRQRRR). NPs@PEI-TAT combined the advantages of metal nanoparticles and peptides, where the presence of metal nanoparticles effectively mitigated the cytotoxicity associated with cationic molecules, making them suitable for biological applications. The NPs@PEI-TAT complexes were incubated with DNA, leading to the formation of Au@PEI-TAT/DNA and Ag@PEI-TAT/DNA (100–200 nm in diameter), which can deliver DNA into the nucleus of stem cells while maintaining high cell viability. Consequently, the metal NPs@PEI-TAT complexes demonstrated the efficient genetic manipulation of epidermal stem cells, inducing their neural differentiation.

Photothermal therapy (PTT) is a promising technique that utilizes light-converting agents to induce heat and “burn” cancer cells. In a recent study by Pan et al. [48], a strategy for intranuclear PTT was developed. This was achieved by delivering gold nanorods (GNRs-NLS) with a size of approximately 10.5 × 40.5 nm to the nucleus via conjugation with TAT peptide (YGRKKRRQRRRC) (Figure 6). The GNRs-NLS showed enhanced intracellular uptake and intranuclear delivery. The study employed a very low level of near-infrared irradiation, at 0.2 W/cm^2^, which is significantly lower than the maximum permissible exposure of skin. Unlike conventional PTT, this low level of irradiation did not cause cell necrosis. Tumors treated with GNRs-NLS under this ultralow NIR irradiation exhibited significant regressions. This report demonstrated that intranuclear PTT using GNRs-NLS greatly mitigated side effects and improved biosafety, making it a promising approach for cancer therapy.

### 3.2. Polymeric Micelles

Polymeric micelles are small colloidal particles with a size typically ranging from 10 to 100 nm [49]. These micelles are formed through the self-assembly of amphiphilic block copolymers, which consist of one hydrophilic unit and one hydrophobic unit. The self-assembly process occurs when the concentration of the block copolymer is above the critical micelle concentration (CMC). The hydrophobic portion of the block copolymer forms the core of the micelle, while the hydrophilic portion forms the shell in an aqueous environment. This unique structure allows for the encapsulation of hydrophobic drugs within the core of the micelle, while the hydrophilic shell stabilizes the micelle in solution and prevents aggregation [50,51].

Polymeric micelles possess various advantages that make them ideal for drug delivery in cancer treatment. Firstly, they can improve the solubility of hydrophobic or poorly water-soluble drugs, thereby enhancing their bioavailability. Secondly, their small size allows for extended circulation in the blood by evading the mononuclear phagocytic system, while also being large enough to avoid fast renal clearance. The high structural stability of polymeric micelles allows for drug retention and stability upon dilution in the body. Furthermore, they can be targeted to specific tissues by modifying their surface with various ligands using different surface chemistries. Additional benefits include reduced side effects, easy and reproducible scale-up, ability to slow down opsonization, and longer circulation times when hydrophilic moieties such as PEG are incorporated to provide an effective steric barrier [50,51,52].

The active targeting of nanoparticles (NPs) has emerged as a strategy to enhance the therapeutic potential of NP-based drugs by facilitating their delivery to specific cellular locations. This is achieved by attaching ligands that specifically recognize the target cells onto the surface of NPs, such as micelles. The ligands can selectively bind to receptors or proteins that are overexpressed on the surface of the target cells, promoting the accumulation of NPs in these cells. Different types of ligands, including peptides, antibodies, aptamers, and small molecules, can be used depending on the target cells and the desired therapeutic effect. Overall, the active targeting of NPs has shown promise in preclinical and clinical studies as a way to improve the delivery and efficacy of drug therapies.

Yu et al. [53] developed a drug delivery system for cancer therapy by creating dual-decorated polymeric micelles that target the nucleus of cancer cells. The micelles were decorated with folic acid (FA) and NLS peptide (Ac-CGYGPKKKRKVGG) to target folate receptor (FR)-positive cancer cells. The micelles were made using cholesterol-modified glycol chitosan (CHGC) that was conjugated with FA and NLS peptide to form NFCHGC micelles. These micelles were loaded with the anticancer drug doxorubicin (DOX) to create DOX/NFCHGC with a diameter of 257 nm. The study found that NFCHGC micelles were efficient in intracellular trafficking, including endosomal/lysosomal escape and nucleus transportation in FR-positive KB cells. DOX/NFCHGC showed stronger cytotoxicity against KB cells compared to other DOX formulations. Blank polymeric micelles displayed low toxicity and good biocompatibility in vivo. The study conducted by Yu et al. indicates that the dual-decorated polymeric micelles they developed have promising potential for cancer treatment, particularly for targeting FR-positive tumor cells and delivering drugs to the nucleus [53].

Hoang et al. [54] designed 31 nm block copolymer micelles (BCMs) labeled with indium-111 (^111^In) and the HER2-specific antibodies trastuzumab (Herceptin, TmAb-Fab) and NLS peptide (CGYGPKKKRKVGG) to directly target the nucleus of HER2-overexpressing breast cancer cells. These micelles, called ^111^In/NLS_2_-TmAb-Fab-BCMs, were investigated in mice with subcutaneous BT-474 and MDA-MB-231 xenografts expressing high and low levels of HER2, respectively, for their pharmacokinetics, biodistribution, tumor uptake, and intratumoral distribution. The TmAb-Fab fragments on the micelle surface facilitated the binding and internalization of BCMs by HER2-positive breast cancer cells, and NLS conferred nuclear localization capability. Targeting BCMs resulted in a 5-fold increase in tumor uptake in HER2-overexpressing BT-474 tumors, as well as a greater level of cellular uptake and nuclear localization compared to non-targeted formulations. Overall, this study presents a promising platform for the effective delivery of chemo- and/or radiotherapy in vivo, providing a potential strategy for the treatment of HER2-overexpressing breast cancer [54].

Wang et al. developed a highly efficient nucleus-targeted co-delivery vector capable of delivering p53 genes and doxorubicin (DOX) directly into the nucleus of cancer cells [55] (Figure 7). The vector, called TAT (YGRKKRRQRRRC)-conjugated chitosan poly-(N-3-carbobenzyloxy-lysine) (CPCL) (TAT-CPCL), was designed with a surface modification of TAT-chitosan. When loaded with p53 and DOX (TAT-CPCL/p53/DOX), the resulting nanoparticles had a diameter of 80–110 nm. Importantly, TAT-CPCL exhibited significantly enhanced nuclear entry compared to CPCL alone, leading to high gene transfection efficiency, increased apoptosis, and reduced cell viability in HeLa cells. The promising results indicate the potential of TAT-CPCL as a vector for cancer gene therapy and suggest its use as a template for designing improved co-delivery systems.

### 3.3. Dendrimers

Dendrimers, which are highly branched, nanoscale macromolecules with diameters ranging from 1 to 20 nm, have shown great potential as drug delivery candidates [56]. Dendrimers differ from traditional linear polymers due to their mono-dispersity, high symmetricity, and surface polyvalency. The repeated growth reactions involved in dendrimer synthesis result in increased generation and branching, eventually forming a three-dimensional spherical structure [57,58].

Dendrimers can be synthesized using two main methods: convergent [59,60] and divergent [61]. The divergent method was developed by Fritz Vogtle and his team in 1978 and involves progressive polymerization from the core to the periphery [62]. On the other hand, the convergent method, introduced by Hawker and Frechet in 1988, involves polymerization from the periphery to the core [63]. Dendrimers have a well-defined core–shell architecture and low polydispersity, meaning they have a uniform size distribution. The synthetic process allows control over dendrimer characteristics such as size, surface charge, peripheral functional groups, and solubility. For example, higher-generation dendrimers have a larger size, a larger interior cavity, and more terminal functional groups [58].

Polyamidoamine (PAMAM) dendrimers have demonstrated superior biocompatibility and a larger nucleic acid loading capacity compared to branched polyethylenimine (PEI). The spheroidal shape, nanoscale size, and cationic surface of PAMAM dendrimers also aid in the cellular uptake of the nucleic acid complexes [58].

Lee et al. [64] developed a dendrimer called RKRARH-PAMAM G2, which incorporates NLS sequences derived from human papillomavirus type 11 E2 protein (RKRAR) conjugated to low-generation PAMAM dendrimer generation 2 (PAMAM G2). Additionally, histidine was introduced to reduce cytotoxicity. Histidine and its derivatives have gained recognition as natural antioxidants because the imidazole moiety present in these compounds is known for its ability to effectively scavenge hydroxyl radicals. In Neuro-2A and NIH3T3 cell lines, RKRARH-PAMAM G2 exhibited high transfection efficiencies and low cytotoxicity compared to polyethyleneimine 25 kDa (PEI 25 kDa). Previous studies by Lee et al. [65,66,67] also showed that incorporating NLS sequences from Herpesviridae or Influenza B Virus Nucleoprotein, or fibroblast growth factor 3, into PAMAM G2 dendrimers can enhance gene delivery and transfection efficiency in multiple cell lines. Cooper et al. [68] conducted a study which revealed that a stable three-part polyplex comprising PAMAM dendrimer G4, NLS (DDDDDDVKRKKKP), and plasmid DNA could enter the nucleus more effectively. Their research has shown that the transfection efficiency is largely influenced by the ratio of G4: NLS: plasmid. Specifically, the polyplex prepared with a ratio of 1:60:1 for G4:NLS:pGFP demonstrated significantly higher transfection efficiency compared to the control group (G4/pGFP, 0.5:1) [68].

It is widely acknowledged that the hydrophobic drugs can be accommodated in the interior hydrophobic cavities of dendrimers. Wu et al. [69] developed a dendrimer-based drug delivery system that is able to deliver the drug to the nucleus by using NLS. The dendrimer was modified with a specific ligand called PGM (PKKKRKV-GFLG-Mp), which contains a nuclear localization sequence (PKKKRKV), enzyme-sensitive tetrapeptide (Gly-Phe-Leu-Gly, GFLG), and pH-sensitive molecule morpholine (Mp), using maleimide active polyethylene glycol ester (NHS-PEG-MAL) to create PAMAM-PEG-PGM. Doxorubicin (DOX) was then loaded into the cavity of PAMAM to create DOX/PAMAM-PEG-PGM. In vitro release studies showed that DOX release from DOX/PAMAM-PEG-PGM nanoparticles occurred in a pH- and enzyme-triggered manner. The nanoparticles were able to promote cell internalization through the charge switching of morpholine and achieve nuclear internalization through the mediation of a composite formed by NLS and importin α/β receptor. In vivo studies using H22 tumor-bearing BALB/c mice as a model showed that the nanoparticles could preferentially accumulate in the tumor site and had a tumor inhibition rate of 88.47%. Overall, these nanoparticles have the potential to be used as a promising nucleus-targeting therapy for enhancing antitumor activity [69].

In one study, Wang et al. introduced a novel delivery carrier called TAT-SS-PAMAM-D3, which was designed to specifically target the nucleus while offering the ability to adjust its size [70] (Figure 8). The surface of the carrier is functionalized with TAT peptide (YGRKKRRQRRRC), and a disulfide linkage is incorporated between D2 and D3. The TAT-SS-PAMAM-D3/p53 complex, with a diameter ranging from 35 to 60 nm, successfully enters the nucleus and exhibits high gene transfection efficiency, promoting apoptosis while reducing cell viability in HeLa cells. This TAT-functionalized carrier with disulfide linking shows promise as a potential vector for cancer gene therapy, offering a novel approach to the development of improved gene delivery systems.

Despite much progress, the clinical translation of dendrimer-based drug delivery systems has been limited due to their biocompatibility and toxicity which are closely linked to the size and surface chemistry of the dendrimer. For example, PAMAM dendrimers of generation 6 and above have high costs and severe toxicity [71], therefore, the higher generations of PAMAM dendrimers are rarely used. The interaction of cationic dendrimers with negatively charged cell membranes can result in the destabilization of biological membranes and thus cause cell lysis, and cationic dendrimers generally exhibit higher toxicity, especially at high doses, than neutral or anionic dendrimers [59,72,73]. Dendrimers have been shown to exhibit high affinity for metal ions, lipids, bile salts, proteins, and nucleic acids, resulting in the disruption of biological processes and leading to toxicity [74]. In addition, the difficulty and expense associated with dendrimer synthesis need to be addressed before clinical translation can be achieved [75].

### 3.4. Mesoporous Silica Nanoparticles

Among various nanocarriers, mesoporous silica nanoparticles (MSNs) have exhibited exceptional characteristics and versatility, consequently finding extensive applications in the field of nanomedicine delivery in recent years. Owing to their large surface area and tunable pore sizes, MSNs enable highly efficient drug loading [76]. The high drug loading capacity allows for the incorporation of a significant amount of small-molecule drugs. Additionally, a prominent advantage of MSNs lies in their ability to modulate drug release by adjusting the pore size on the surface, resulting in sustained and controlled drug delivery over an extended period [77]. It is well-known that controllable drug release during the entire delivery process plays a critical role in therapeutic efficacy. Moreover, the surface of MSNs can be tailored by introducing targeting ligands or functional groups to achieve site-specific drug delivery. These surface modifications facilitate the selective binding of particles to specific cells or tissues, thereby enhancing drug accumulation at the desired site while minimizing off-target effects. This feature holds promise for MSNs in the context of nucleus-targeted drug delivery [78]. Furthermore, MSNs allow for the co-delivery of hydrophobic and hydrophilic drugs within the same particle, accommodating drugs with different solubilities. Notably, MSNs are generally considered biocompatible and can be engineered for biodegradability. Biodegradable silica-based materials can be designed to degrade gradually, ensuring the elimination of particles from the body without causing long-term adverse effects [79]. This characteristic offers potential for the clinical translation of MSNs as nanocarriers for drug delivery. Numerous researchers have made significant contributions to the development of MSN-based nanomedicine delivery systems in previous studies.

Pan et al. pioneered the development of a cell-nucleus-targeted delivery system by conjugating TAT peptide with mesoporous silica nanoparticles (MSNs) [80] (Figure 9). This innovative design allowed for the efficient loading of a substantial amount of the anticancer drug doxorubicin (DOX) into the MSNs-TAT conjugates (pore size 2–3 nm), forming DOX@MSNs-TAT. The presence of TAT peptide on the surface of MSNs facilitated the binding of the nanoparticles with importin α and β, enabling their entry into the cell nucleus through the nuclear pore complex. As a result, DOX was directly released into the cell nucleus, achieving targeted nanomedicine delivery to the nucleus. The experimental results showed that MSNs-TAT nanoparticles with a size range of 25–50 nm exhibited efficient importation into the cell nucleus. In contrast, MSNs-TAT nanoparticles with sizes of 67 and 105 nm were predominantly located in the cytoplasm and the perinuclear region after the same incubation duration. The targeted delivery of DOX to the cell nucleus significantly enhanced the therapeutic efficacy of this anticancer drug.

The TAT peptide, characterized by a highly cationic motif, has been widely utilized for the cellular uptake and intracellular delivery of various cargoes. Nonetheless, it is important to highlight that this peptide lacks specificity for tumor cells, as it can bind to the cell membranes of both cancerous and normal cells. Hence, to ensure the precise delivery of nanoparticles to tumor cells, it is critical to block TAT peptides prior to the release of TAT-labeled nanoparticles into the cell cytosol. To block the TAT peptide, Zhao et al. developed a novel multifunctional nano-drug delivery system based on the reversal of peptide charge [81] (Figure 10). This nanomedicine delivery system utilized mesoporous silica nanoparticles (MSNs) with a diameter of approximately 35 nm and pore size of 2.2 nm as carriers. The researchers first bound MSNs with TAT-FITC, followed by the binding of citraconic anhydride (Cit) and YSA-BHQ1. The anticancer drug doxorubicin (DOX) was then loaded into the MSN complex (MSN/COOH/TAT-FITC/Cit/YSA-BHQ1/DOX, with a diameter of 50 nm). TAT-FITC was utilized for efficient intranuclear delivery, while the YSA-BHQ1 quencher group specifically targeted the tumor EphA2 membrane receptor [81]. The system operated based on the instability of Cit under acidic conditions, resulting in the restoration of positive charge on the TAT peptide and the repulsion of the positively charged YSA segment [82]. After removing the YSA-BHQ1 fragment, the fluorescence of FITC was restored, and the TAT peptide guided the nanoparticles to bind with the nuclear transport receptors importin α and β, enabling entry into the cell nucleus through the nuclear pore complex (NPC). This nanocarrier system remained stable at physiological pH, rapidly released the drug in acidic buffer, and was readily absorbed by MCF-7 cells [81]. The experimental results demonstrated the promising prospects of this nanomedicine delivery system for both the diagnosis and treatment of cancer cells.

Li et al. conducted a study to develop a novel nucleus-targeted system using mesoporous silica nanoparticles (MSN) [83] (Figure 11). The main objective of this system was to specifically target cancer stem cells (CSCs) and facilitate the delivery of therapeutic drugs to the nucleus. To achieve this, the researchers performed the surface modification of the MSN with antibody against CD133, a specific marker for CSCs. Additionally, they incorporated TAT peptides (YGRKKRRQRRR) that could be thermally triggered under an alternating magnetic field (AMF). The resulting labeled MSN, known as CD133/TAT/TPZ-Fe_3_O_4_@mSiO_2_, exhibited a diameter of 91 nm (the diameter of the Fe_3_O_4_@mSiO_2_ nanoparticle is 50 nm and the pore diameter is 2.8 nm). Tirapazamine (TPZ) is an anticancer drug that demonstrates remarkable selectivity in targeting hypoxic cancer stem cells (CSCs). TPZ exerts its effects within the nucleus by interacting with reductases, leading to the generation of a transient oxidizing radical. The targeted delivery of drugs to the nucleus resulted in the induction of apoptosis in CSCs through a combination of thermotherapy and hypoxia-activated chemotherapy. In vivo experiments demonstrated that the nucleus-targeted nano delivery system effectively suppressed tumor growth with minimal side effects. Molecular mechanism studies revealed that the system achieved CSC elimination by inhibiting the hypoxia signaling pathway. This innovative nucleus-targeted nano delivery system holds great promise for the development of efficient platforms for CSC-targeted cancer therapy.

### 3.5. Liposomes

Liposomes are a class of spherical structures (usually 50–500 nm in diameter) composed of one or more lipid bilayers that encapsulate various drugs within their aqueous and lipid layers [84,85]. Their structural similarity to cell membranes makes them attractive candidates as nano-transport carriers. Hydrophilic drugs are encapsulated in the hydrophilic core of the liposome, while hydrophobic drugs are incorporated into the lipid membrane. Amphiphilic or neutral drugs can be encapsulated on the phospholipid layer of the aqueous membrane [86]. The properties of liposomes, such as their excellent biocompatibility, strong controllability, high targeting ability, and excellent reversibility, make them attractive for drug delivery applications. The relatively large size of liposomes makes it hard for them to enter the nucleus by travelling through the NPC. To deliver drugs to the nucleus by liposome, the general method is using the NLS to label the drugs (usually DNA or antibody) and encapsulating the NLS-labeled drugs in the liposome. The NLS-labeled drugs will be released into cytosol after cell membrane fusion with the liposome and target the nucleus via the NLS to avoid efflux by MDR [87,88,89,90,91].

In one study, researchers developed a delivery vehicle composed of a synthetic NLS peptide derived from the SV40 virus, a luciferase-encoding PGL3 plasmid, and a cationic lipid DOTAP-: DOPE (1:1 *w/w*) liposome. This delivery system was successfully transfected into SKnSH mammalian neuroblastoma cells, resulting in a three-fold increase in luciferase expression compared to cationic liposome controls. By investigating the factors that impede the rate of transgene expression, it is possible to uncover new strategies for improving the efficiency and effectiveness of non-viral gene therapy methods [89].

Similarly, Rosada et al. demonstrated the potential of a cationic liposome loaded with nuclear localization signal (NLS)-labeled DNAhsp65 (referred to as NLS/DNA/cationic liposome) for the treatment of tuberculosis (Figure 12). The targeting peptide consists of 21 amino acid residues (KCRGKVPGKYGKGPKKKRKVC) which encompasses a nuclear localization signal from SV40T (PKKKRKV), a cationic nuclear shuttle sequence (KCRGKVPGKYGKG), and a cysteamide group at the C-terminus. By complexing the peptide with the plasmid DNAhsp65 and incorporating it into cationic liposomes, a pseudo-ternary complex was formed. The pseudo-ternary complex exhibited a controlled size (approximately 250 nm in diameter), a spherical-like shape, and various lamellae within the liposomes, as confirmed by transmission electron microscopy. The insertion of the peptide/DNA into the liposome structure was verified through a fluorescence probe accessibility assay. Furthermore, the addition of the peptide did not induce cytotoxicity in vitro, and the treatment using four times less DNA compared to naked DNA showed similar therapeutic effects against tuberculosis. These findings collectively suggest that the pseudo-ternary complex holds promise as a gene vaccine for the treatment of tuberculosis [87].

In addition to gene therapy, liposomes have been utilized for the delivery of anticancer drugs to the nucleus. In one particular study, a nuclear-targeted delivery system was developed to overcome drug resistance in breast cancer (MCF-7/Adr) by leveraging nucleolin’s active transport to the nucleus and its affinity with aptamers. The drug doxorubicin hydrochloride (Dox·HCl) was inserted into the aptamer AS1411 (Ap-Dox) and encapsulated within the aqueous interior of liposomes (Lip (Ap-Dox)). In vitro investigations demonstrated that upon diffusion of Lip (Ap-Dox) into MCF-7/Adr cells, the Ap-Dox complex strongly bound to nucleolin, which contains a bipartite nuclear localization signal (NLS: KRKKEMAKQKAAPEAKKQKV), leading to its eventual entry into the cell nuclei. By utilizing this drug delivery system, Dox·HCl efficiently accumulated in the nuclei, effectively inducing cancer cell death [89].

Wang et al. engineered a liposome formulation that incorporated a monoclonal antibody called TuBB-9, designed to selectively recognize a biologically active form of the protein Ki-67, which predominantly localizes in the nucleus (Figure 13). To enhance the nuclear targeting of the antibody, they conjugated it with a nuclear localization signal (NLS) derived from the SV-40 virus. This modification allowed the antibody to efficiently localize to the nucleus and exert its therapeutic effect, effectively eliminating tumor cells that expressed Ki-67 [88].

In one study, Torchilin et al. demonstrated the effectiveness of liposomes modified with TAT peptide (referred to as TATp-liposomes) in facilitating rapid and efficient translocation into the cell cytoplasm, followed by migration into the perinuclear zone [92]. In vitro experiments involved transfecting mouse NIH/3T3 fibroblasts and rat H9C2 cardiomyocytes with TATp-liposome-DNA complexes. The transfection efficiency of TATp-liposome-associated pEGFP-N1 plasmid, encoding the green fluorescent protein (GFP), was found to be high, while exhibiting lower cytotoxicity compared to commonly used cationic lipid-based gene delivery systems. Moreover, the intratumoral injection of TATp-liposome-DNA complexes into Lewis lung carcinoma tumors in mice resulted in GFP expression within tumor cells. Collectively, these findings support the notion that TATp-conjugated plasmid-bearing liposomes serve as an efficient DNA delivery system capable of bypassing the lysosomal compartment and delivering their cargo directly into the cytoplasm and the vicinity of the nuclei. This transfection system holds promise for various cell treatment protocols in vitro or ex vivo, as well as localized gene therapy in vivo.

## 4. Summary and Perspective

A summary of nanoparticles and their applications in nuclear drug delivery is depicted in Figure 14. Classical nuclear localization signals (cNLSs) are primarily utilized for labeling polymeric micelles and dendrimers, with no reports of their application in labeling mesoporous silica nanoparticles and liposomes. In contrast, the TAT peptide has been employed for labeling all the mentioned nanoparticles and facilitating drug delivery to the nucleus. Metal nanoparticles are mainly used for delivering DNA to the nucleus. Polymeric micelles are employed for delivering doxorubicin (DOX), DNA, and Indium-111 to the nucleus, while dendrimers are utilized for delivering DOX and DNA. Mesoporous silica nanoparticles are primarily used for delivering DOX and tirapazamine (TPZ), which can fit into the nanoparticle pores and reach the nucleus. Liposomes are employed for delivering DNA and antibodies to the nucleus, either through direct TAT labeling or by carrying cNLS-labeled DNA and antibodies. Among the various nanoparticles, DNA is the most delivered cargo to the nucleus, followed by DOX. Therefore, nuclear drug delivery using nanoparticles proves to be highly useful for delivering DNA and DOX to the nucleus.

In the context of therapeutic gene delivery, non-viral systems are expected to play a significant role. Although they have traditionally exhibited lower efficiency in gene expression compared to viral delivery systems, non-viral systems offer several advantages such as simplified production, reduced toxicity, and no risk of infection. However, a limitation of many non-viral systems is the lack of a mechanism for efficient gene transport into the nucleus. To overcome this challenge, the incorporation of nuclear localization signal (NLS) peptides can combine the enhanced expression capabilities of viral delivery systems with the safety and ease of preparation associated with non-viral delivery systems. By integrating NLS peptides, non-viral systems can improve the targeting and transport of therapeutic genes to the nucleus, thereby enhancing their overall effectiveness as gene delivery vehicles.

Classical nuclear localization signals (cNLSs) rely on importin α and importin β to facilitate the import of cargo into the nucleus. This process is unidirectional and enables the accumulation of cargo in the nucleus. Moreover, cNLSs offer specific nuclear targeting capabilities for delivering cargo. However, the maximum size of nanoparticles that can traverse the nuclear pore complex (NPC) using cNLSs is approximately 40 nm in diameter. Depending on the physical properties of the cargo, larger cargo (>40 nm diameter) can potentially be transported through the NPC with the aid of flexible shapes during transport. However, for rigid-core nanoparticles such as metal nanoparticles or mesoporous silica nanoparticles (MSNs), which do not change shape during transportation, cNLSs cannot facilitate their transport if the core diameter exceeds 40 nm.

In contrast to cNLSs, the TAT peptide can facilitate the import of nanoparticles with larger and rigid cores (>40 nm in diameter) into the nucleus. The exact mechanism behind this phenomenon is still not fully understood. Therefore, it is important to investigate how these rigid and large nanoparticles traverse the nuclear envelope after being labeled with TAT. One possible explanation is that in live cells, TAT-labeled nanoparticles indirectly bind to nuclear components during cell division, allowing the large cargo to remain inside the nucleus. To test this hypothesis, the use of digitonin-permeabilized cells is essential, because cell entry is not a concern and the intact nuclear envelope prevents cargoes from binding to nuclear components. The other explanation is that the TAT peptide can penetrate the nuclear envelope and allow large cargo to enter the nucleus and bypass the nuclear pore complexes (NPCs) (Figure 3). Compared to cNLSs, the TAT peptide, acting as both a cell-penetrating peptide (CPP) and nuclear localization signal (NLS), exhibits greater efficiency in delivering nanoparticles to the nucleus. The higher efficiency of transporting cargo across the cell membrane [93,94], rather than enhanced nuclear transport, is likely the primary factor contributing to the improved effectiveness of TAT peptide in delivering cargo to the nucleus. Additionally, TAT peptide has been shown to also target mitochondria [95], indicating its potential to deliver multiple drugs to different organelles such as the mitochondria and nucleus where they can exert their effects. However, for the accurate nuclear delivery of drugs, it is crucial to test the targeting specificity of the TAT peptide.

The efficiency of importing NLS-labeled nanoparticles depends on the density of NLS labeling [20,47,96], highlighting the significant impact of optimal labeling density on import efficiency. The digitonin-permeabilized cell can serve as a simple and straightforward transport system to test the effect of different labeling densities on nanoparticle import efficiency. To gain molecular insights into the import of TAT- and cNLS-labeled nanoparticles, techniques such as single-molecule tracking and super-resolution microscopy can be highly valuable. Single-molecule tracking has been employed to study the transport of importin and different cargos [96,97], while photoactivated localization microscopy (PALM) has provided high-resolution structures of FG-nucleoporins (FG-nups) [98]. Furthermore, MINFLUX nanoscopy enables the tracking of individual molecules with nanometer-level localization precision for extended periods of time [99]. These techniques can shed light on the passage of TAT- or cNLS-labeled nanoparticles through the cell membrane, their escape from endo/lysosomes, cytoplasmic trafficking, and their entry into the nucleus. This comprehensive understanding can serve as a powerful platform for investigating different labeling methods, labeling densities, and types of functionalization, and their impact on the fate of nanoparticles.

## Figures and Tables

**Figure 1 cells-12-01637-f001:**
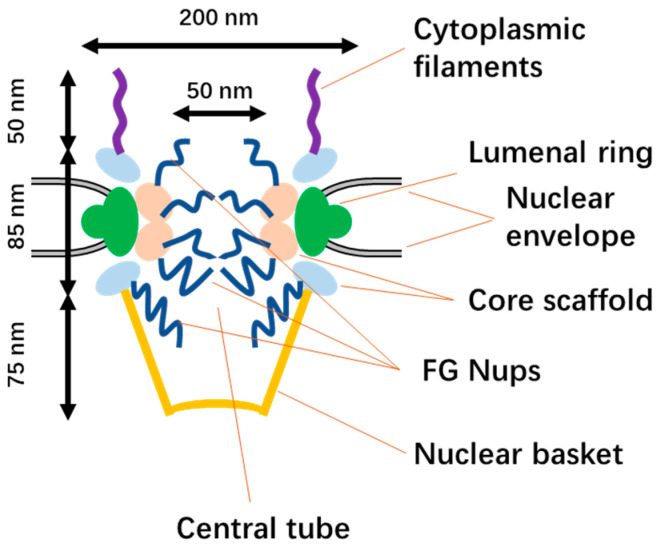
The structure of the NPC.

**Figure 2 cells-12-01637-f002:**
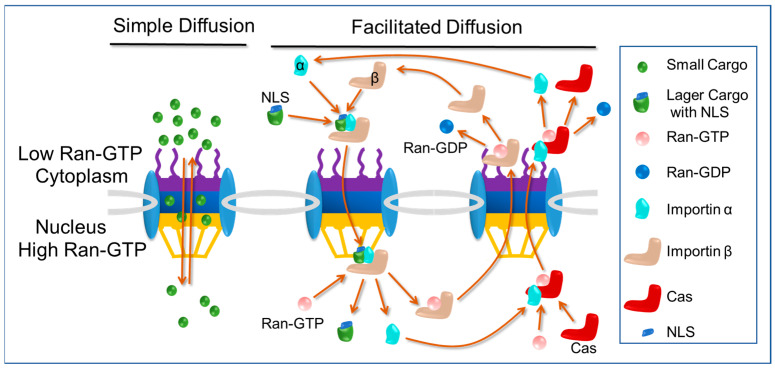
Free diffusion and classical nuclear import pathway.

**Figure 3 cells-12-01637-f003:**
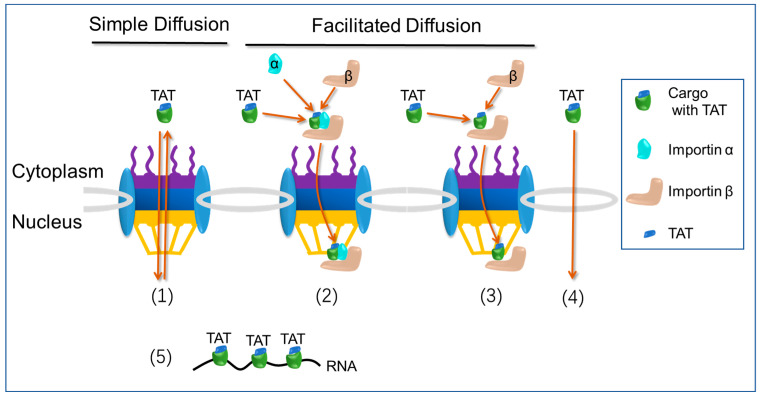
The transactivator of transcription (TAT) nuclear import pathway.

**Figure 4 cells-12-01637-f004:**
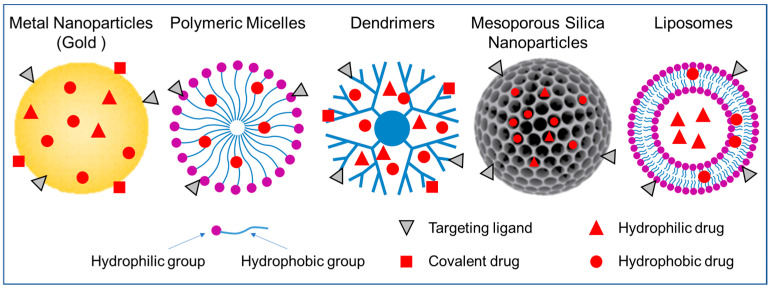
Summary of the nanoparticles used for NLS labeling.

**Figure 5 cells-12-01637-f005:**
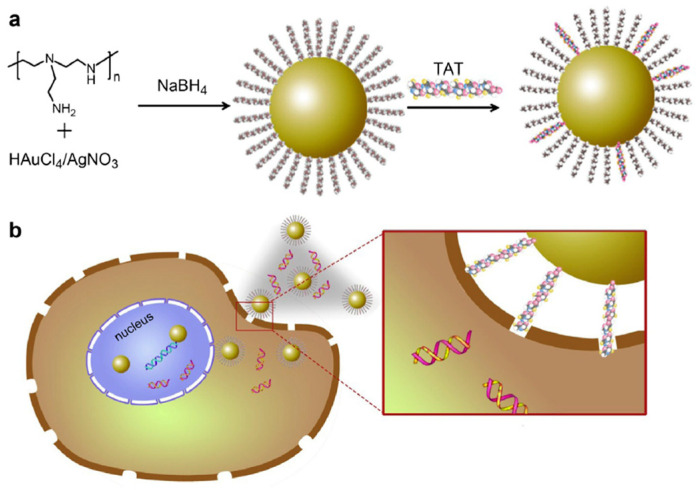
(**a**) Scheme of the preparation of TAT peptide-conjugated cationic noble metal nanoparticles. (**b**) Schematic illustration of positively charged TAT peptide-conjugated noble metal NPs binding with DNA for intercellular delivery. Reprinted (adapted) with permission from [47]. Copyright 2014 Elsevier Ltd.

**Figure 6 cells-12-01637-f006:**
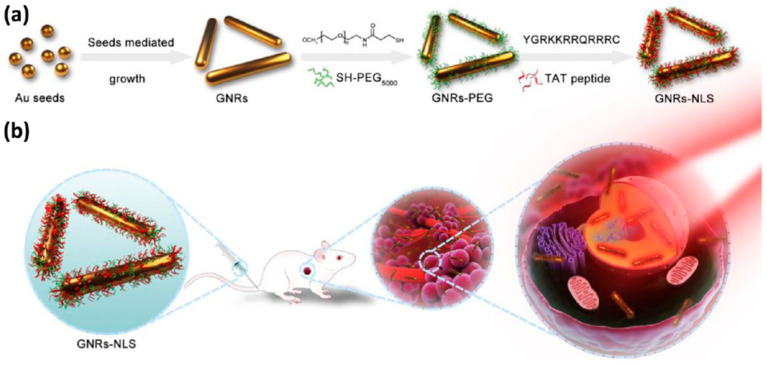
(**a**) Schematic illustration of the synthetic procedure of GNRs-NLS. (**b**) Schematic representation of nuclear-targeted photothermal therapy of GNRs-NLS. Reprinted (adapted) with permission from [48]. Copyright 2017 American Chemical Society.

**Figure 7 cells-12-01637-f007:**
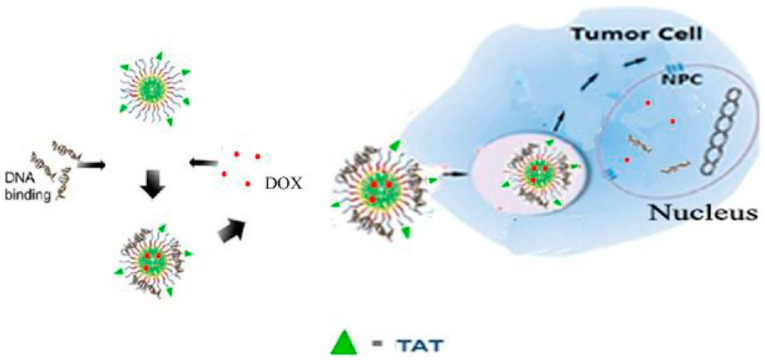
A high-efficiency nucleus-targeted co-delivery vector that delivers genes and drugs directly into the nucleus. Reprinted (adapted) with permission from [55]. Copyright 2018 Elsevier Ltd.

**Figure 8 cells-12-01637-f008:**
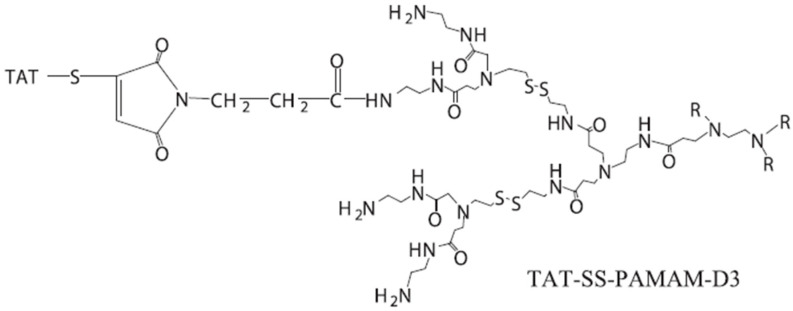
Synthesis of the TAT-SS-PAMAM-D3. Reprinted (adapted) with permission from [70]. Copyright 2018 Elsevier Ltd.

**Figure 9 cells-12-01637-f009:**
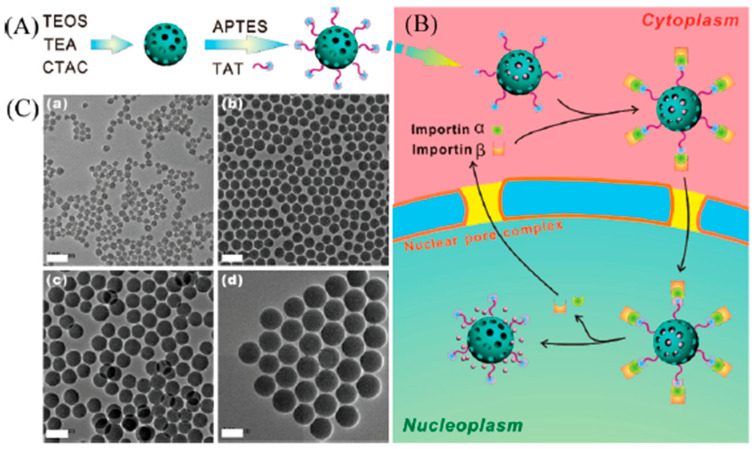
(**A**) Schematic diagram of the procedures for preparing amine-group-conjugated and TAT-C6-FITC peptide-conjugated MSNs. (**B**) Schematic illustration of transport of DOX@MSNs-TAT across the nuclear membrane. (**C**) TEM images of MSNs with sizes of (**a**) 25, (**b**) 50, (**c**) 67, and (**d**) 105 nm. Scale bars: 100 nm. Reprinted (adapted) with permission from [80]. Copyright 2012 American Chemical Society.

**Figure 10 cells-12-01637-f010:**
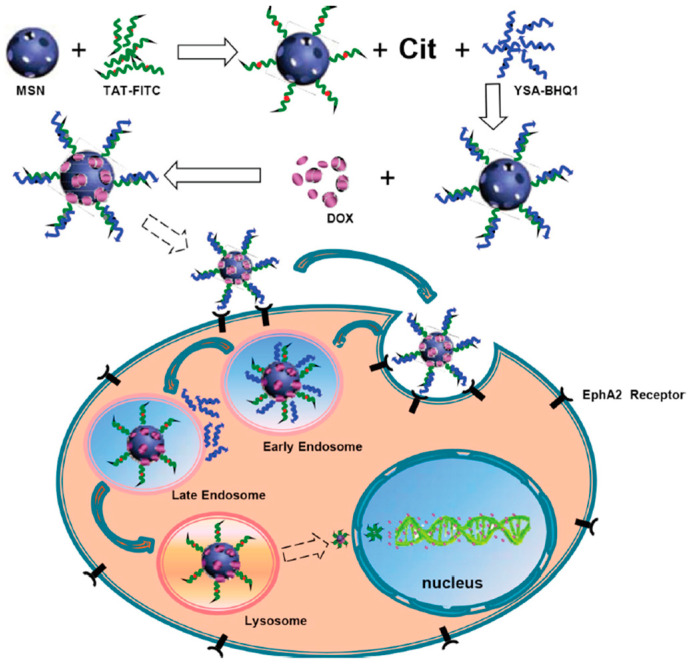
Illustration of the synthesis and mechanism of action in MCF-7 cells of MSN/COOH/TAT-FITC/Cit/YSA-BHQ1/DOX. Reprinted (adapted) from [81]. No permission required.

**Figure 11 cells-12-01637-f011:**
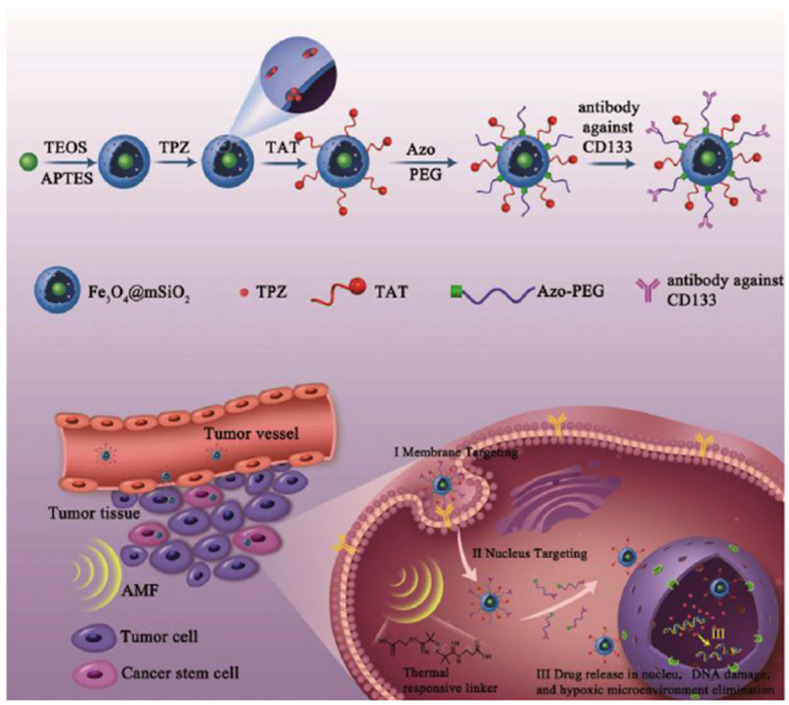
Fabrication of antibody against CD133/TAT/TPZ-Fe_3_O_4_@mSiO_2_ (abbreviated: CD133/TAT/TPZ-Fe_3_O_4_@mSiO_2_) NPs and schematic of the multistage targeting strategy for cancer stem-cell-targeting therapy. Reprinted (adapted) with permission from [83]. Copyright 2019 Elsevier Ltd.

**Figure 12 cells-12-01637-f012:**
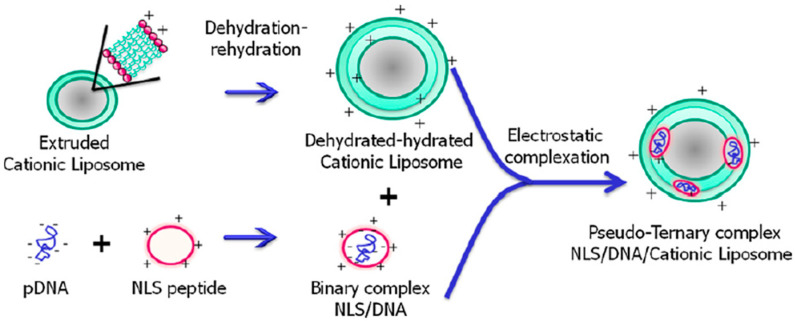
Schematic representation of the pseudoternary complex. Reprinted (adapted) with permission from [87]. Copyright 2012 Elsevier Ltd.

**Figure 13 cells-12-01637-f013:**
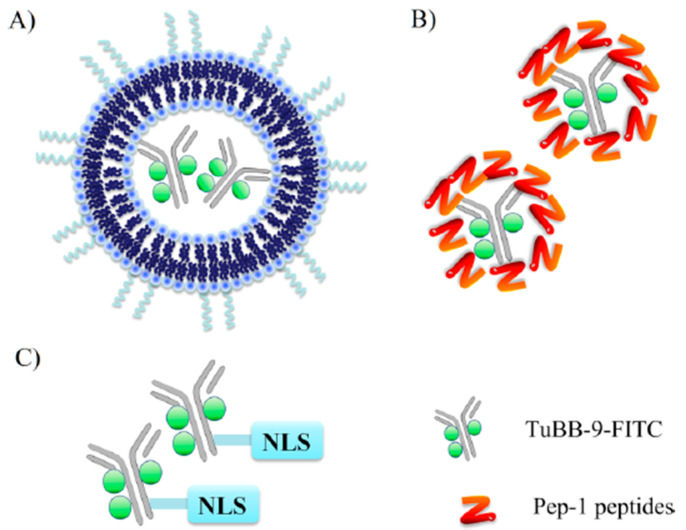
Diagram of the different antibody constructs used for targeting the Ki-67 protein. (**A**) TuBB-9-FITC encapsulated in liposomes. (**B**) Pep-1 sequences noncovalently bound to TuBB-9-FITC. (**C**) TuBB-9-FITC conjugated to NLS by a SMCC linker. Reprinted (adapted) with permission from [88]. Copyright 2015 American Chemical Society.

**Figure 14 cells-12-01637-f014:**
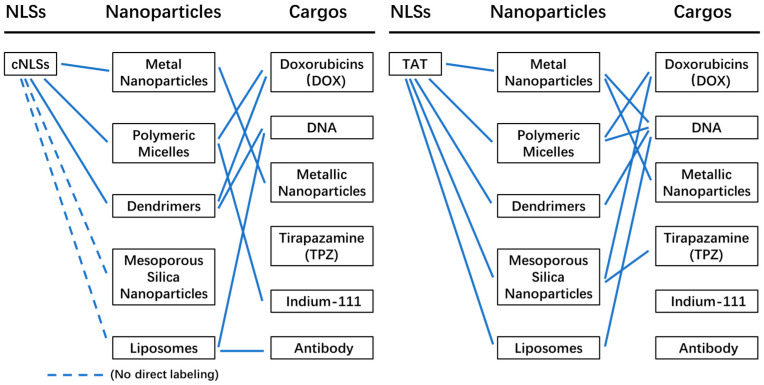
Summary of nanoparticles and their applications in nuclear drug delivery.

## Data Availability

Not applicable.

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
