# Peer review of "Nuclear Delivery of Nanoparticle-Based Drug Delivery Systems by Nuclear Localization Signals"

_cells, 2023, doi:10.3390/cells12121637_

Round 1

Reviewer 1 Report

The paper of Nie et al covers a very interesting and emerging topic such is the delivery of nanoparticles to the cell nuclei. Prior to its publication, authors must address the following remarks:

Major revision: 

-  Point 2.3. Authors should develop more this point as it is crucial for the design of these drug delivery systems.

- Point 3.4 and 3.5 Why liposomes and polymer drugs are included if there is not any study on the use of NLS? Why these ones and not other drug delivery systems? Authors should reconsider including this part and focus and develop more on other drug delivery systems using this strategy. 

-Point 4. The perspectives that the authors give are not really related to the nuclear delivery of nanoparticles. This part should be improved.

- The authors should discuss in depth (maybe in a new point) the impact of this strategy for gene therapy.

- Figure 4. I do not find it useful in the way that it does not provide any interesting information. Authors should improve it.

Minor revision:

- Line 72. Authors talk about nanometers but they should correlate it with the molecular weight.

- Line 85. Rephrase the sentence.

-Line 115. Gold or golden?

-Line 144. Correct Typo

-line 157. Which peptide was better at the end?

-Line 187. Is the term "burn cancer cells " suitable?

- Line 229. The nanoparticles display a size of 250 nm. How did they reach the nucleus ? The size seems too big (according to what you claim in the previous points). Please, explain better this.

Line 277. What is the role of histidine and how they were introduced?

Line 307. " and higher"?

Line 341. Encapsulated in or on?

Line 351. I would not say the term "enormous" given the data provided.

-

Author Response

We really appreciate the valuable feedback provided by you, and we have carefully addressed all your comments and suggestions during the revision of the manuscript. We have thoroughly incorporated the necessary changes as recommended by the editor and reviewers. All the revisions made to the manuscript have been clearly marked by underlining them (track changes).

Reviewer 2 Report

This work summarizes nucleus-targeted nanoparticle-based drug delivery systems by using NLS. The manuscript is correctly written and the review has an appropriate structure. However, the paper shows some inconsistences that preclude the acceptance in its present form. Below I introduce some comments that can help to improve the final level of this manuscript.

1) There is no mention in the text of the benefits of using these systems, the advantages of directing them towards the nucleus, or the drawbacks of requiring access to the nucleus in order to discharge the drug.

2) Section 3, which is entitled "Nanoparticles for Nucleus Targeting" and is the most crucial section, requires a thorough evaluation. The authors have mentioned five types of nanoparticles: golden nanoparticles, polymeric micelles, dendrimers, polymer drug conjugates, and liposomes. However, dedicating a whole section to polymer drug conjugate and liposomes, only to reveal that no current publications exist, does not appear to be logical. They assume that they will be good for nucleus targeting due to the characteristics of these materials.

3) The revision needs to incorporate papers discussing metallic nanoparticles or mesoporous silica particles, because as a review it should include a diversity of materials. For instance (but not limited to), here I have included some of them:

·        Biomaterials 2019, 200, 1–14. Mesoporous silica nanoparticles

·        Chemical Engineering Journal 2020, 380, 122458. TID nanoparticles

·        ACS Appl. Mater. Interfaces 2023, 15, 10541−10553. Manganese Dioxide Nanoparticles

·        Adv. Healthcare Mater. 2017, 6, 1601289. Multifunctional Magnetic Nanoparticles

·        Angew. Chem. Int. Ed. 2022, 61, e202201486. PtIV Nanoparticles

·        PNAS, 2020, 117, 2771. (NLS)-conjugated polymersome nanocarriers

4) To make Section 3 more engaging, it is necessary to include additional figures or tables.

The language is quite good. A revision by a native speaker will probably improve the quality of the text.

Author Response

(The authors gave the same response as above.)

Round 2

Reviewer 1 Report

My decision is: accept in present form.

Reviewer 2 Report

The authors have acomplished all points risen by the reviewers, the manuscript is now ready for acceptance.